# Genome-Wide Identification, Expression and Stress Analysis of the GRAS Gene Family in *Phoebe bournei*

**DOI:** 10.3390/plants12102048

**Published:** 2023-05-21

**Authors:** Jiarui Chang, Dunjin Fan, Shuoxian Lan, Shengze Cheng, Shipin Chen, Yuling Lin, Shijiang Cao

**Affiliations:** 1International College, Fujian Agriculture and Forestry University, Fuzhou 350002, China; cherichang@163.com; 2College of Forestry, Fujian Agriculture and Forestry University, Fuzhou 350002, China; fandunjin@foxmail.com (D.F.); csz231619@163.com (S.C.); fjcsp@126.com (S.C.); 3College of Horticulture, Fujian Agriculture and Forestry University, Fuzhou 350002, China; 4Key Laboratory of Fujian Universities for Stress Physiology Ecology and Molecular Biology of Forest, Fuzhou 350002, China

**Keywords:** GRAS gene, *Phoebe bournei*, phylogenetic analysis, abiotic stress, expression analysis

## Abstract

GRAS genes are important transcriptional regulators in plants that govern plant growth and development through enhancing plant hormones, biosynthesis, and signaling pathways. Drought and other abiotic factors may influence the defenses and growth of *Phoebe bournei*, which is a superb timber source for the construction industry and building exquisite furniture. Although genome-wide identification of the GRAS gene family has been completed in many species, that of most woody plants, particularly *P. bournei*, has not yet begun. We performed a genome-wide investigation of 56 *PbGRAS* genes, which are unequally distributed across 12 chromosomes. They are divided into nine subclades. Furthermore, these 56 *PbGRAS* genes have a substantial number of components related to abiotic stress responses or phytohormone transmission. Analysis using qRT-PCR showed that the expression of four *PbGRAS* genes, namely *PbGRAS7*, *PbGRAS10*, *PbGRAS14* and *PbGRAS16*, was differentially increased in response to drought, salt and temperature stresses, respectively. We hypothesize that they may help *P. bournei* to successfully resist harsh environmental disturbances. In this work, we conducted a comprehensive survey of the GRAS gene family in *P. bournei* plants, and the results provide an extensive and preliminary resource for further clarification of the molecular mechanisms of the GRAS gene family in *P. bournei* in response to abiotic stresses and forestry improvement.

## 1. Introduction

Throughout the life of a plant, temperature, water and salt stresses may recur over time and gradually affect its normal physiological activities. For example, salinity can cause oxidation, ionization, and osmosis, whereas heat and dryness can damage chloroplast membranes and promote chlorophyll breakdown, lowering photosynthetic efficiency [1]. These abiotic pressures may become more common in the natural environment as a result of climate change. Transcription factors (TFs) can precisely control the transcription of target genes via conjugation to deoxyribonucleic acid (DNA) sequences in response to unfavorable conditions [2,3,4]. TFs can help us understand the adaptive response networks that comprise many biological processes.

GRAS is an extremely significant TF that can regulate a variety of physiological processes in higher plants [5]. The GRAS gene family is named after the first three members to be discovered: Gibberellin Acid Insensitive (GAI), Repressor GA1 (RGA), and Scarecrow (SCR) [6,7]. The GRAS structural domain of a protein consists of an extremely volatile amino (N-) terminus region and a highly conserved carboxyl (C-) terminus region [8]. Leucine heptapeptide repeat I (LHRI), leucine heptapeptide repeat II (LHRII), VHIID, PFYRE, and SAW motifs make up the C terminus, while the N terminus is a disordered domain [9,10]. The main factors thought to influence the taxonomy of GRAS families are the sequences’, structures’, and phylogenetic relationships’ diversity. Depending on the plant species tested, the number of subfamilies within the GRAS gene family can vary (DELLA, PAT1, LISCL, HAM, SCR, SHR, LAS, SCL3 in model plants and NSP1, NSP2, RAD1, SCLA, SCLB, RAM1, DL, SCL4/7, SCL32 in other plants) [10,11,12,13]. Take *Castanea mollissima*, for instance—the GRAS family is segmented into nine subfamilies, namely DELLA, PAT1, SCL3, SHR, SCR, LISCL, HAM, LAS and DLT [14]. The 150 GRAS genes in *G. hirsutum* were divided into 15 subfamilies: SCR (10), DLT (4), OS19 (4), LAS (2), SCL4/7 (4), OS4 (4), OS43 (2), DELLA (8), PAT1 (27), SHR (28), HAM (24), SCL3 (9), LISCL (20) and G_GRAS (4) [6]. The author also analyzed the evolution of algae and found that the GRASs originated from the period between algae and moss [6]. In the woody plant *Betula alba*, 265 GRAS genes were sorted into 17 groups using the analysis of 40 birch genes, 32 *A. thaliana* genes, 38 *O*. *sativa* genes, 52 tea tree genes, 59 *P. dactylifera* genes, and 44 *T. cocoa* genes [15]. Research such as this shows that within the plant kingdom, this family of genes is incredibly diverse. Additionally, various subfamilies may have distinct purposes during plant development due to variations in conserved structural domains. The SCR and SHR subfamilies operate in root and leaf growth—for example, SHR and SCR regulate bundle sheath cell development in *A. thaliana* and are also involved in asymmetric cell division and radial patterns in *A. thaliana* roots [16,17]. A recent study found that SCR-regulated SHR cells can autonomously inhibit formative pericyte division in moss plant leaf vein formation [18]. These results suggested that these genes are involved in regulating the direction of cell division and establishing genetic regulatory networks in land plants, both in flowering plants and bryophytes [18]. SCL is an essential subfamily involved in root cell elongation in the plant *A. thalian* by combining numerous signals and responses to different signaling/abiotic stresses (cold) as a co-activator in *G. hirsutum* [6,19]. These results suggest that the SCL subfamily may affect the regulation of plant signaling molecules. Meristem integrity in petunia stems is preserved by the HAM subfamily [20]. Moreover, the DELLA subfamily has different roles in the signaling of phytohormones such as gibberellins [21,22,23], which induce flower, stem and root growth in *A. thaliana* by regulating GA signaling [17]. In addition, DELLA genes also dominate plant tissue expansion [23,24]. Furthermore, similarly to the LAS subfamily’s function affecting leaf and stem growth in *G. hirsutum*, the GV6 subfamily in *Solanum lycopersicum* also plays an essential role in the elongation of stems [6,25].

The GRAS family in various species of plants has been under extensive study in recent years, and not just at a surface level. In response to abiotic conditions such as drought, salt, and photo-oxidative stimuli, GRAS has been proven to be instrumental in numerous investigations [13]. Hydrogen peroxide (H_2_O_2_) accumulation is decreased in transgenic *O*. *sativa* due to OsGRAS23’s favorable regulation of drought tolerance and oxidative stress tolerance by manipulating the activity of genes that engage in responding to stress [26]. GRAS1 in tobacco is implicated in signal transduction pathways that increase reactive oxygen species (ROS) levels in response to various stress treatments, suggesting a role in transcriptional regulation in response to stress [27,28]. Overexpression of the grape (*Vitis amurensis*) GRAS protein VaPAT1 enhanced abiotic stress tolerance in transgenic *A. thaliana* [29]. *Populus tremula*’s tolerance to drought and salt was improved by overexpressing the GRAS protein SCL7 [30]. Birch (*Betula alba*) demonstrated a salt-tolerant phenotype upon the overexpression of BpGRAS34, as well as increased ROS scavenging capabilities and proline content [15].

*Phoebe bournei* (Hemsl.) Yang, a member of the family Lauraceae, is one of the most economically and ecologically valuable timber species [31]. Below 1000 m in altitude, *Phoebe bournei* prefers to live in evergreen broad-leaved forests with healthy and well-drained soils [32]. Because of its straight trunk, large and dense crown, beautiful shape, excellent material and resistance to decay, fine grain structure, fragrant wood, and smooth and beautiful chipping, it is widely considered to be one of the best timber species for building construction, furniture making, craft carving, and ornamental purposes [33,34]. As major players in the molecular mechanisms of plant stress resistance, GRAS family members are responsible for a wide diversity of physiological procedures, covering root formation, fruit development, hormone signaling, stem cell maintenance, initiation of the axillary meristem in response to light, generation of male gametophytes, and response to adversity [17,20,35,36,37]. Despite the fact that genome-wide investigations of GRAS genes have been completed for a number of botanical species, no such analysis has been undertaken for *P. bournei*, and the role of GRAS proteins in abiotic strain responses remains unclear. In this work, we described 56 GRAS genes after conducting a genome-wide analysis of this gene family in *P. bournei*. Quantitative RT-PCR (qRT-PCR) was employed to examine the effects of exposure to drought, salt, and temperature stressors on a sub-set of *PbGRASs* genes.

In this research, we utilized the chromosome-level genomic information of *P. bournei* to systematically identify its GRAS family genes and analyzed the physicochemical nature of proteins and the structure and chromosomal distribution of the genes and established the evolutionary relationships, promoter cis-acting elements, and epistatic expression patterns of drought, salt and temperature stress genes in an attempt to offer a frame of reference for further research on the biological functions of *P. bournei* GRAS genes in the process of stress resistance. This research will be fundamental in elucidating the underlying molecular mechanisms of GRAS responsiveness to drought, salt and temperature stress, and will supply quality genes for genetically engineered breeding in *P. bournei* for improved resistance to either drought, salinity or high and cool temperatures.

## 2. Results

### 2.1. Phylogenetic Analysis of PbGRAS Gene

On the basis of the updated genome assembly, we identified 150 presumptive GRAS genes: 56 in *P. bournei*, 34 in *A. thaliana*, and 60 in *O. sativa*. The evolutionary interrelationships of the proteins of PbGRAS were structured using phylogenetic trees (Figure 1). Phylogenetic analysis revealed that these 150 GRAS proteins can be separated into nine groups (PAT1, SHR, LISCL, HAM, SCL4/7, LAS, SCL3, DELLA, and SCR). These results, as with those published for birch (*Betula platyphylla*), *Dendrobium chrysotoxum*, and soybean (*Glycine max*), demonstrated some variability in the number of GRAS genes in different subfamilies [15,38,39]. PbGRAS proteins are unevenly distributed in nine subfamilies, most of which belong to the DELLA subfamily (18 members), followed by the PAT1 subfamily (12 members). Only seven, six, five, three, two and one PbGRAS protein were detected in the HAM, SCR, SHR, SCL3, LISCL and LAS subfamilies, respectively. Several of the PbGRAS proteins were shown to have strong phylogenetic relationships with proteins from other species (bootstrap support ≥ 80), which could indicate that they are similar to GRAS proteins from *A. thaliana* or *O. sativa* serving the same purpose.

### 2.2. Chromosome Distribution and Synthesis Analysis of PbGRAS Gene

A total of 56 PbGRAS proteins were retrieved from the *P. bournei* genome databases and were characterized. Their physico-chemical properties were investigated further with the help of ExPasy (Table 1). The majority of these proteins have classical GRAS structural domains with around 400–700 amino acids (aa), reaching up to 777, while PbGRAS5 has a severely truncated GRAS domain with only 119 amino acids. The 56 PbGRASs had molecular weights ranging from 13.682 kDa to 84.836 kDa. Theoretical pI values were between 4.86 and 6.9 for the majority of PbGRAS proteins, with only a few pI values in the range of 7 to 9. This finding suggests that the majority of GRAS proteins are negatively charged, and only a minority are positively charged. The grand average of hydropathicity index for all PbGRAS proteins (from −0.555 to −0.014) indicates that all PbGRAS proteins are hydrophilic; these results are similar to those of GRAS proteins in *Betula platyphylla* [15]. The majority of the forecasted instability indices were above 40 (from 40.08 to 61.97), demonstrating that most PbGRAS proteins are unstable except for PbGRAS5 (39.37), PbGRAS35 (39.67), PbGRAS37 (35.3), and PbGRAS54 (32.34). Most of the PbGRASs do not have introns, suggesting that the sequence of PbGRASs is, to some extent, conserved.

The 56 *PbGRASs* identified were further localized on 12 *Phoebe bournei* chromosomes (Chr01 to Chr12) (Figure 2). Overall, the distribution of these 56 *PbGRASs* on the 12 chromosomes was heterogeneous. The distribution of *PbGRASs* on *P. bournei* chromosomes was variable and heterogeneous among chromosomal regions. No *PbGRASs* were detected on Chr06. Among the above chromosomes, the highest distribution of *PbGRASs* was on Chr08, with 10 *PbGRASs* (17.9%), followed by Chr07 (9, 16.1%), then Chr01 and Chr02 (6, 10.7%). Five *PbGRASs* (8.9%) were located on Chr05, four *PbGRASs* (7.1%) were located on Chr03, 04, and 09, three *PbGRASs* (5.3%) were located on Chr10 and 12, and two *PbGRASs* (3.6%) were located on Chr11. Based on previous studies, we hypothesized that no significant association or correlation was found between the amount of GRAS genes and chromosome length [5,40].

Tandem duplication events are defined as chromosomal regions that contain two or more genes within 200 kb and perform a critical task in the further generation of new functions in gene families [41]. Five tandem replication events were identified on Chr07 and Chr08, including *PbGRAS1*/*PbGRAS47*, *PbGRAS3*/*PbGRAS17*, *PbGRAS5*/*PbGRAS36*, *PbGRAS48*/*PbGRAS3*, and *PbGRAS53*/*PbGRAS37*, involving a total of nine *PbGRASs* (Figure 2). Except for *PbGRAS1* and *PbGRAS47* from the SCR subfamily, and *PbGRAS5* and *PbGRAS36* from the PAT1 subfamily, all genes that participate in tandem replication events belong to the DELLA subfamily, implying that the DELLA cluster, as the largest subfamily, plays a substantial role in the growing number of GRASs [7,42]. In addition, there are six fragment replicates situated at or near the terminal end of chromosomes 1, 2, 3, 4, 5, 7 and 10. To further investigate the gene duplications, we constructed a comparative tandem map of the three *PbGRAS* families (Figure 3). Among them, the intensity of association with *PbGRAS* genes was ranked from highest to lowest as follows: *O*. *sativa* (22), *A. thaliana* (18). The linear homology plots suggest that tandem replication events are the main mechanism for gene family expansion in plants, and more importantly, for gene family diversity [43,44].

### 2.3. Conserved Structural Domain Motif Analysis of PbGRASs

As a further exploration of the GRAS sequence characteristics and structure of *P. bournei*, we optimized the conserved motifs and intronic structures of *P. bournei* (Figure 4). To examine the structural specifics of the PbGRAS protein, we estimated 10 conserved motifs in the MEME program. GRAS proteins from the same subfamily contain comparable motif combinations in general, implying that GRAS proteins of the same subfamily likely have identical functions. Nearly all GRAS proteins included Motifs 1, 2, 3, 4, 5, 6, 7 and 10, demonstrating that the motifs are highly conserved and might have significant effects within the GRAS family. Motif 8 is only distributed in the PAT1 subfamily; Motif 9 is missing in the HAM subfamily except for *PbGRAS15*. *PbGRAS5* contains only Motif 3 and Motif 7, *PbGRAS23* contains only Motifs 2, 4, 6 and 7, and *PbGRAS43* contains only Motifs 1, 4, 5 and 7. Some motifs are distributed only in specific positions of the patterns. Motif 7, for instance, is consistently dispersed near the conclusion of the sequence, but Motif 5 is usually disseminated at the beginning. The intended uses of the majority of these preserved motifs are unknown. The variances in the order of appearance of these motifs across subfamilies imply that GRAS proteins from various subdivisions may have distinct purposes. Additionally, multiple genes within the same subfamily exhibit distinct motif distributions, implying that the tasks they perform may be unique as well. The tendency for similar motifs to cluster together in specific subgroups of GRAS proteins suggests the possibility of cross-functional similarities between these proteins. In addition, to determine the differences in gene structures, we compared the exon and intron structures of 56 *P. bournei* GRAS genes. Most of the GRAS genes have minimal or no introns, which was also evidenced in the present study with 67.86% of *PbGRAS* genes having no introns (Figure 4 and Table 1). Five and three introns were detected in *PbGRAS18* and *PbGRAS29*, respectively. This suggests that the family was relatively conservative in evolution.

### 2.4. Multiple Sequence Alignment and Cis-Acting Element Analysis of PbGRASs

Multiple sequence alignment of PbGRAS proteins showed a degree of conservation in the GRAS gene family. According to the output scores of multiple sequence alignment, there is a high similarity in the 320–360 and 460–510 regions, and it can be inferred that the function of these regions may be similar (Figure 5). Cis-acting components can affect the transcribed state of their linked genes since they are non-coding DNA regions in gene promoters. The occurrence of cis-acting elements in promoters might be related to the variety of gene-related processes and expression trends. We submitted base pair sequences from the 2 kb region upstream of the *PbGRAS* gene promoter to the PlantCARE database and found 19 cis-acting parts linked to reactions to environmental stress or phytohormone transduction, including nine stress-responsive elements and five phytohormone-related elements (Figure 6). As demonstrated in the picture, all *PbGRAS* genes have between seven and 11 cis-acting elements. Each GRAS gene has at a minimum one element associated with a strain response such as light responsiveness, essential for anaerobic induction, defense and stress responsiveness, low-temperature responsiveness, drought inducibility, anoxic specific inducibility, circadian control, and wound responsiveness, having a major effect in the generation of strain responses. Meanwhile, five phytohormone-related elements in the *PbGRAS* genes appear in most phytohormones; these include abscisic acid responsiveness, MeJA responsiveness, gibberellin responsiveness, auxin responsiveness, and salicylic acid responsiveness. Among them, we discovered that the photosensitive component is the richest cis-acting component, as it is extensively represented in the *PbGRAS* promoter region. These findings suggest that *PbGRASs* might be tolerant to non-biological stresses, such as light, salt, cold, and drought, and may be involved in plant evolutionary and developmental processes.

### 2.5. Heat Map of PbGRAS Gene Expression in Different Tissues

For analyzing the variation in *PbGRAS* exposure modes, we incorporated gene epistasis data from five tissues: leaf, root xylem, stem xylem, root bark, and stem bark (Figure 7). The outcomes displayed that *PbGRAS7* and *PbGRAS14* had the highest expression in the root xylem and stem xylem, and moderate expression in the root bark and stem bark; in particular, *PbGRAS7* reached maximum expression in stem xylem, which indicated that the performance of *PbGRA7* expression was closely related to the cultivation of the stem xylem. In addition, *PbGRAS10*, *PbGRAS41*, and *PbGRAS16* genes were also highly expressed in these four tissues, but their expression was lower in the root xylem and stem xylem compared with *PbGRAS7* and *PbGRAS14*. Meanwhile, the expression of *PbGRAS10* was comparatively higher in the root and stem bark. This is similar to the results in pepper, where members of the PAT1 subfamily, such as *CaGRAS30* and *CaGRAS34*, are expressed at higher concentrations in the stems than in other organs [3].

### 2.6. Abiotic Stress Experiments on the GRAS Gene Family of P. bournei

To investigate the responses of GRAS genes to abiotic stress, we tested the expression responsiveness of *P. bournei* genes to drought, salt, and temperature stress (Figure 8). The relatively high expression of *PbGRAS7*, *PbGRAS10*, *PbGRAS14* and *PbGRAS16* in different tissues can be seen more clearly in the gene expression heat map. In addition, cis-acting element analysis showed that *PbGRAS7*, *PbGRAS10*, *PbGRAS14* and *PbGRAS16* contain a large number of elements associated with drought and temperature stresses. Thus, we selected these four genes to validate the results further. The findings reveal that the levels of manifestation of GRAS genes were modulated in response to dryness, salt, and temperature. The expression of *PbGRAS10*, *PbGRAS14*, and *PbGRAS16* did not change significantly after 4 h of immersion with a nutrient solution containing 10% PEG, and only the expression of *PbGRAS7* was enhanced and was significantly higher at 6 h, about three times that of the control (Figure 8a). *PbGRAS7* may have a non-significant effect on *P. bournei*’s osmotic stress. When treated with 10% NaCl (Figure 8b), we discovered that *PbGRAS7* and *PbGRAS14* genes were repressed to different degrees. The expression of *PbGRAS16* increased somewhat after 4 h, while *PbGRAS10* increased significantly at 4 h after treatment and increased about 30-fold after 24 h compared with the expression at the beginning of treatment. One may presume that *PbGRAS7* may have a significant action in the tolerance of *P. bournei* to drought strain. Under the 10 °C treatment (Figure 8c), the expression of all genes increased to different degrees after 12 h. The expression of *PbGRAS10* and *PbGRAS16* was already increased at 4 h, and *PbGRAS7*, *PbGRAS14*, and *PbGRAS16* still maintained high expression levels after 24 h. The expression of *PbGRAS10* and *PbGRAS16* was increased at the beginning of the treatment. Among them, *PbGRAS10* expression was remarkably enhanced after 12 h and was about 25-fold compared with the control group. Under 40 °C treatment (Figure 8d), the expression of every gene was increased to various extents and remained high after 24 h. In particular, *PbGRAS10* and *PbGRAS16* expression increased about 5-fold after 24 h of treatment compared with the CK group. It can be postulated that *PbGRAS10* and *PbGRAS16* have significant effects on the plant’s reactions to temperature stress.

## 3. Discussion

Plants primarily use gene regulation as an adaptive mechanism in response to environmental stressors [45]. The growth, differentiation, and stress responses of tissues and organs are tightly controlled by GRAS proteins [46]. The subtropical broadleaf evergreen tree *P. bournei* is extremely important, both economically and ecologically. Water scarcity and temperature variation severely affect *P. bournei* growth and wood accumulation. Jin and others found that *P. bournei* can regulate its own protective enzyme system and osmotic substances to adapt to drought under drought stress [47]. Wang showed that polyamines can increase soluble sugar and chlorophyll content under drought and improve the drought resistance of *P. bournei* [48]. However, there is a dearth of literature on the regulatory signaling pathways that help *P. bournei* to endure abiotic challenges such as drought. As GRAS genes have such a critical function in the signaling of *P. bournei*, it is crucial to comprehend their expression pattern. The correct identification of abiotic tolerance genes in *P. bournei* will be aided by a fundamental and thorough bioinformatic investigation of them.

Many plant species have been found to contain GRAS family genes, with 34 GRAS gene members in *A. thaliana* [17], 60 in *O*. *sativa* (*Oryza sativa*) [11], and 153 in wheat (*Triticum aestivum*) [49]. We undertook a genome-wide investigation of the GRAS gene family in *P. bournei* and identified 56 GRAS proteins, the majority of which have characteristic GRAS structural domains totaling roughly 360 aa, most of which share a conserved C-terminus (Table 1 and Figure 5), which is consistent with the study by Liu and Wang [50]. Most of the conserved sequences appeared in the 270–610 region. Given that intron loss in eukaryotes is massive due to selection pressure and that the GRAS group is a retrotransposon group that accelerates the generation of intron-less genes from bacterial to eukaryotic gene sets through both horizontal gene transfer and gene dimerization [51,52,53], it is possible that large and frequent gene duplication events have led to changes in the number of *PbGRAS* genes (38 out of 56, Table 1). This investigation found five simultaneous duplication events including a total of nine *PbGRASs*, and the majority of the genes implicated in these events belonged to the DELLA subfamily (Figure 2). However, some *PbGRASs* evolved different exon–intron outcomes, which may indicate that they evolved adaptive functions in order to adapt to their environment. Moreover, most GRAS proteins in Table 1 have pI values located at around 5~7, with a smaller possibility of interacting nonspecifically with acidic proteins [54], and the GRAS set may be engaged in protein–protein interfaces [55]. The findings of chromosomal localization reveal that the 56 *PbGRASs* identified were unevenly located on 12 chromosomes of *P. bournei* (Figure 2), with the highest distribution being on Chr08, accounting for 17.9% of the *PbGRASs*, while no *PbGRASs* were detected on Chr06. These outcomes are analogous to other research, such as where no *MdGRAS* geneswere found on chromosome 16 of *Malus domestica*, while chromosome 11 contained the most, accounting for 15.75% of the *MdGRAS* genes [56]. It is possible to attribute this to evolutionary processes such as gene duplication and chromosomal transfer. Chr08 has the largest amount of GRAS genes and may be one of the critical chromosomes for plants to deal with abiotic pressures. The addition of GRAS genes from *A. thaliana* and *O*. *sativa* expanded the dataset to 150 GRAS proteins divided into nine subfamilies after phylogenetic analysis (Figure 1). However, the conserved motifs in different subfamilies are different, which may be due to the subfamilies of PbGRASs performing different biological functions in the growth of *P. bournei*, as the motif arrangement was mostly similar among PbGRAS proteins of the same subfamily, suggesting that PbGRAS proteins of the same subfamily may have similar functions. As shown by To and Wang et al., GRAS proteins are stochastically assigned in the phylogenetic tree [7,57], but individuals from the same subfamily tend to perform comparable tasks [50].

Characterization of cis-acting elements demonstrated that *PbGRAS* genes may participate in multiple biotic procedures by regulating a variety of target genes associated with hormone induction, evolution, metabolism, and abiotic stresses (Figure 6). Among these cis-acting elements, those correlated to drought and temperature stress such as drought inducibility and low-temperature responsiveness are assigned to the promoter regions of *PbGRAS* genes, such as *PbGRAS10* and *PbGRAS14*. There are many elements in *PbGRAS14* that react to drought stress and salt stress, but fewer light response elements compared with other *PbGRAS* genes, which indicates that this gene may be less involved in light response and is the main gene used to cope with salt and drought stress. By comparing the other genes of the DELLA subfamily, there were relatively few cis-acting elements for light response, but more cis-acting elements for stress response, which indicated that the DELLA gene family may play a role in the stress response of *P. bournei*. Then, focusing on the stress experiment (Figure 8), it could be seen that the expression of *PbGRAS14* changed significantly under NaCl and 40 °C stress, which is also consistent with our analysis results. A similar example is the case for *CmGRAS* genes, which are more closely related to the reference species in Chinese chestnut and were found to have more cis-elements and showed higher expression modes at different phases of buds and fertile or aborted ovules, suggesting their significant roles in the evolution and functioning of the *CmGRAS* gene family [14]. Additionally, the results can be seen more clearly in the expression heat map of the genes; *PbGRAS7* and *PbGRAS14* are highly expressed in the root xylem and stem xylem, while *PbGRAS10* is relatively highly expressed in the root bark and stem bark. This likely relates to the fact that GRAS genes display a differential expression pattern across different tissues [13]. Consistent with some other observations, such as the expression of *NnuGRAS-05*, *NnuGRAS-10*, and *NnuGRAS-25* in lotus root bark compared to their expression in the leaf and petiole, members of the PAT1 subfamily seem to be more highly expressed in the root than in the leaf [58]. The expression levels of *CaGRAS30* and *CaGRAS34* in pepper were higher in stems than in other organs [3]. Almost all PAT1 subfamily members were highly expressed in eggplant stems [25]. This may be due to the fact that PAT1 subfamily members help to maintain rootstock growth and resist environmental disturbances.

Combining these findings and analysis, we selected four genes (*PbGRAS7*, *PbGRAS10*, *PbGRAS14*, and *PbGRAS16*) for validation of the stress treatments. By subjecting abiotic stress-treated *P. bournei* seedlings to qRT-PCR analysis, it was found that drought, salt and temperature stress conditions significantly induced the expression of the four *PbGRASs* in *P. bournei* plants (Figure 8). In particular, the expression of *PbGRAS10* significantly surged 30-fold and 25-fold after salt treatment and low-temperature treatment, respectively, *PbGRAS7* was expressed at a three times higher level than in the control after PEG treatment, and *PbGRAS10* and *PbGRAS16* increased approximately five-fold after high-temperature treatment compared with the control, which was consistent with the outcomes of cis-acting meta-analyses (Figure 6). In contrast, a low-temperature-responsive cis-element was predicted at the *PbGRAS14* promoter, but its expression level was comparable to the control condition. This, to some extent, reflects the fact that the accuracy of the cis-acting element prediction results needs to be further improved. Drought, salt, and temperature stress conditions significantly induced the expression of *PbGRASs* in *P. bournei* plants, suggesting that *PbGRASs* may have some degree of association with stress response. There have been other studies describing similar findings. In *O. sativa* (*Oryza sativa*), *OsGRAS23* in the SCL subfamily is highly sensitive to salinity stress, and the expression of this gene is up-regulated under drought stress [26,59]. Similar results were found in *Halostachys caspica*, where the expression of *HcSCL13* in the SCL subfamily was up-regulated under drought and salt treatments [30]. Salt stress in poplar (*Populus euphratica*) also regulates a stress response transcription factor, *PeSCL7* [60]. This provides more evidence that the SCL subfamily is crucial for adapting to adverse conditions. The GRAS family of grapevine is also regulated by abiotic stresses, for example, in that drought treatment upregulates the expression of *VviPAT3*, *VviPAT4*, *VviPAT6* and *VViPAT7* in seedlings [61], while salt treatment has an effect on *VviRGA5* in the DELLA subfamily [62]. Furthermore, the *Lycopersicon esculentum* gene *SlGRAS2*, which is homologous to *VviPAT4*, is highly sensitive to salt stress [51]. According to a recent report, GRAS genes play a key role in the regulation of salt stress [63], and thus, the role of GRAS in promoting plant growth and salt stress response has received widespread attention. However, further in-depth research is required to understand the function of GRAS genes in response to abiotic stressors in a wider range of plant species.

This study not only determined and examined the expression of *PbGRASs* using qRT-PCR, but it also systematically analyzed the GRAS gene family of *P. bournei*, suggesting that *PbGRASs* play an active part in the physiology of *P. bournei*’s resistance to drought, salt, high, and low temperature stresses. This study provides a basis for elucidating the molecular mechanisms of the *P. bournei* GRAS response to drought, salt, and high and low temperature stresses and posits ideas for the future improvement of *P. bournei* using genetic engineering methods.

## 4. Materials and Methods

### 4.1. Genome Data and Plant Material Source

The genome sequence data and annotation information of *P. bournei* were downloaded from the Sequence Archive of China National GeneBank Database (CNSA) with accession number CNP0002030 [64]. Genome sequence files of *A. thaliana* (version: Arabidopsis_thaliana.TAIR10.dna.toplevel.fa.gz, last modified: 17 February 2023 06:57, 35M) and *O. sativa* (version: Osativa_204_v7.0.fa.gz, last modified: 12 January 2014, 110.4M) were acquired from EnsemblPlents (https://plants.ensembl.org, accessed on 31 October 2022) and Phytozome v13 (https://phytozome-next.jgi.doe.gov, accessed on 31 January 2023), respectively. The RNA-seq data from different tissues of *P. bournei* were download from BioProject (https://www.ncbi.nlm.nih.gov/bioproject/, accessed on 12 March 2023) with accession number PRJNA628065. The plant materials were derived from 1-year-old *P. bournei* seedlings cultured in an artificial climate box under different treatments. *P. bournei* seedlings with the same growth potential for 1 year were selected for treatment, and the materials were divided into the control group and stress treatment group, with 30 strains in the treatment group and 3 strains in the control group. Every 2 strains in the treatment group were used as a biological replicate, and 3 groups of biological replicates were set in each time period. After various treatments, *P. bournei* leaf samples were collected and stored in liquid nitrogen at −80 °C for RNA extraction.

### 4.2. Identification and Physical and Chemical Property Analysis

The conserved domain of the *A. thaliana* GRAS gene family was downloaded from PlantTFDB (http://planttfdb.gao-lab.org/, accessed on 31 October 2022). A local BLASTp search was used to compare the conserved domain between *P. bournei* and *A. thaliana* to screen the candidate GRAS genes in *P. bournei* [39]. The repetitive results of the BLASTp search were removed. The identified protein sequences of GRAS genes from *P. bournei* were submitted to the NCBI for BLASTp to perform further searching. To further identify GRAS gene family members, the GRAS conserved domain HMM model (PF03514) was downloaded from the Pfam database (http://pfam.xfam.org/, accessed on 14 April 2023), using HMMER-3.2.1 (http://hmmer.org/download.html, accessed on 14 April 2023) with an e-value <10^-5^ and other parameters by default. After the identification of GRAS genes in *P. bournei*, the online website ExPASy (https://web.expasy.org/prot-param/, accessed on 26 February 2023) was used to analyze the physical and chemical properties of the GRAS proteins that were identified [65].

### 4.3. Chromosomal Distribution and Gene Duplication of PbGRAS Genes

TBtools was used for grepping the location information of the *PbGRAS* genes from the genome (FASTA) file and the annotation (GFF) file of *P. bournei* [66]. The syntenic relationships of *PbGRAS* were determined using MCScanX with default parameters and plotted using Tbtools [67].

### 4.4. Collinearity Analysis of PbGRAS Genes

The syntenic relationships between *PbGRAS* genes and GRAS genes from *A. thaliana* and *O. sativa* were determined by using MCScanX software. Tbtools was used for visualization.

### 4.5. Phylogenetic Analysis

The sequences of GRAS proteins of *P. bournei*, *A. thaliana* and *O. sativa* were aligned using the Muscle program of MEGA11 with default settings for constructing maximum likelihood (bootstrap replications: 1000) phylogenetic tree [68,69]. iTOL (https://itol.embl.de/, accessed on 29 January 2023) was used to improve and beautify the phylogenetic tree.

### 4.6. Analysis of Conserved Motifs and Gene Structures

The protein sequence of *P. bournei* was identified using the online software MEME (http://meme-suite.org/, accessed on 8 March 2023), and the predicted value of the motif number was 10. The multiple GRAS protein sequences alignment was carried out using Jalview software (https://www.jalview.org/, accessed on 30 March 2023). We used the Batch CD-search (https://www.ncbi.nlm.nih.gov/Structure/bwrpsb/bwrpsb.cgi, accessed on 8 March 2023) with default parameters to detect the conserved domains of the PbGRAS protein.

### 4.7. Multiple Sequence Alignment and Promoter Cis-Element Analysis of PbGRAS Genes

Multiple sequence alignment of *PbGRAS* was performed using Jalview software (version: 2.11.2.6). To explore the cis-acting elements in the sequence, we extracted the upstream 2000 bp sequences from the *P. bournei* genome. The online software PlantCARE (https://bioinformatics.psb.ugent.be/webtools/plantcare/html/, accessed on 11 March 2023) was used to analyze the cis-acting regulatory elements in the promoter region of the *PbGRAS* genes. After selection and categorization, the data were visualized by means of TBtools software.

### 4.8. Treatment in Different Tissue

The fragments per kilobase of transcript per million fragments mapped (FPKM) transcriptomic data from five different tissues (leaf, root xylem, stem xylem, root bark and stem bark) were used to construct a expression profiling with TBtools.

Based on results of cis-element analysis and expression profiling of *PbGRAS*, we conducted different treatment. In the drought and salt stress experiment, 3 seedlings in the control group were soaked in distilled water after root washing, while those in the treatment group were soaked in a nutrient solution containing 10% PEG and 10% NaCl, respectively, and cultured in an artificial climate incubator with a temperature of 25 °C and humidity of 75%. The treatment group was sampled at 4 h, 6 h, 8 h, 12 h and 24 h, respectively, and the control group was sampled at 0h. In the temperature treatment (10 °C and 40 °C), the control group was incubated at room temperature, and the treatment group was incubated at 10 °C and 40 °C. The treatment group was sampled at 4 h, 6 h, 8 h, 12 h and 24 h, with 6 strains in each time period. The 40 °C and 10 °C control groups were sampled at 0 h.

### 4.9. qRT-PCR Analysis

An RNA extraction kit (HiPure Plant RNA Mini Kit from Magen, Shanghai, China) was used to extract RNA, while cDNA was synthesized using the Prime Script RT reagent Kit (Perfect Real Time from Takara, Dalian, China). Primer 3.0 software was used to design specific primers in the non-conserved region of the target gene, which were synthesized by Fuzhou Qingbaiwang Biotechnology Company. Real-time fluorescence quantitative analysis was used with cDNA template (1 μL), cDNA template SYBR Premix Ex TaqTM II (10 μL), specific primers (2 μL), and ddH_2_O reaction program (7 μL): 95 °C for 30 s; 95 °C for 5 s; 60 °C for 30 s; 95 °C for 5 s; 60 °C for 60 s; and 30 s at 50 °C, with 40 cycles in total. The internal reference gene was *PbEF1α* (GenBank No. KX682032) [70]. The expression level of the target gene was calculated using the 2−ΔΔCt method, and the quantitative data were analyzed via t test using SPSS26 software. Finally, GraphPad Prism 9.0 was used to construct the graphs.

## 5. Conclusions

In this study, a total of 56 PbGRAS proteins were identified from the *P. bournei* genome and phylogenetically classified into nine subfamilies. Four GRAS genes, *PbGRAS7*, *PbGRAS10*, *PbGRAS14* and *PbGRAS16*, were induced by stress treatments such as drought, salt, and low and high temperature. The expression of each of the four genes varied under different treatments. These results suggest that *PbGRASs* may play a role in improving the tolerance of transgenic *P. bournei* to drought, salt, and high- and low-temperature stresses. This study lays the foundation for the further elucidation of the functions of *PbGRAS* family members, provides valuable information for further studies on the role of GRAS family genes in the development of abiotic stress resistance in *P. bournei*, and provides a basis for the improvement of *P. bournei*.

## Figures and Tables

**Figure 1 plants-12-02048-f001:**
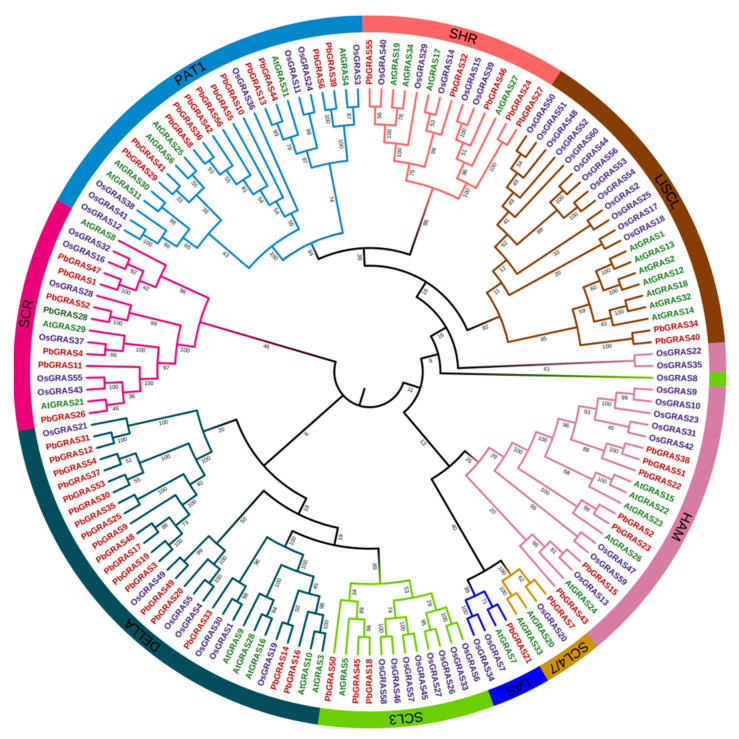
Phylogenetic tree of PbGRAS, AtGRAS and OsGRAS proteins. The different-colored arcs indicate subfamilies of the GRAS family. The tree was constructed using the 56 PbGRAs identified in *P. bournei*, the 34 AtGRAs identified in *A. thaliana* and the 60 OsGRAs identified in *O. sativa* by using MEGA11 with the bootstrap 1000 times.

**Figure 2 plants-12-02048-f002:**
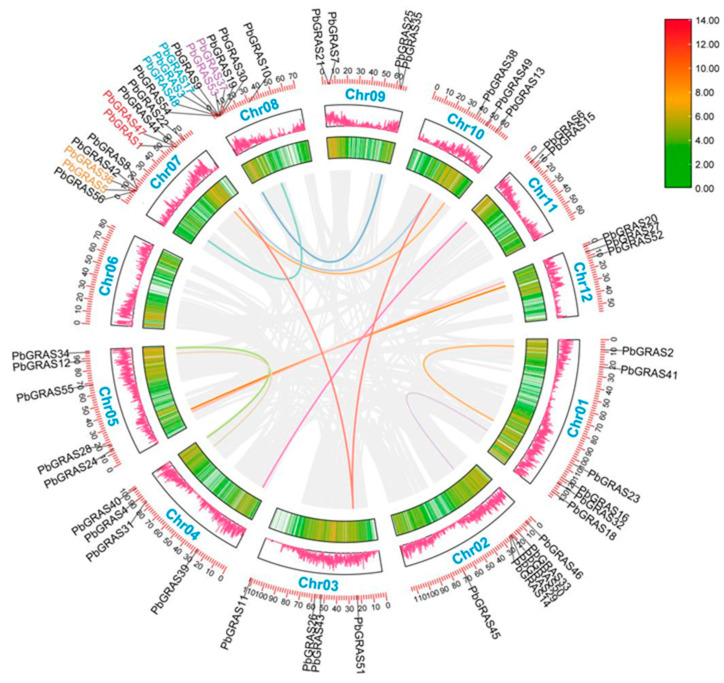
Genomic positions, duplication events and syntenic relationships of *PbGRAS* genes. The green and red blocks in the middle represent the gene density of each chromosome. Those genomes deriving from segmental replication events are linked by colored lines, and the gray lines denote synthetic blocks in the *P. bournei* genome. The genes in purple, blue, red, and yellow are those with tandem replication.

**Figure 3 plants-12-02048-f003:**
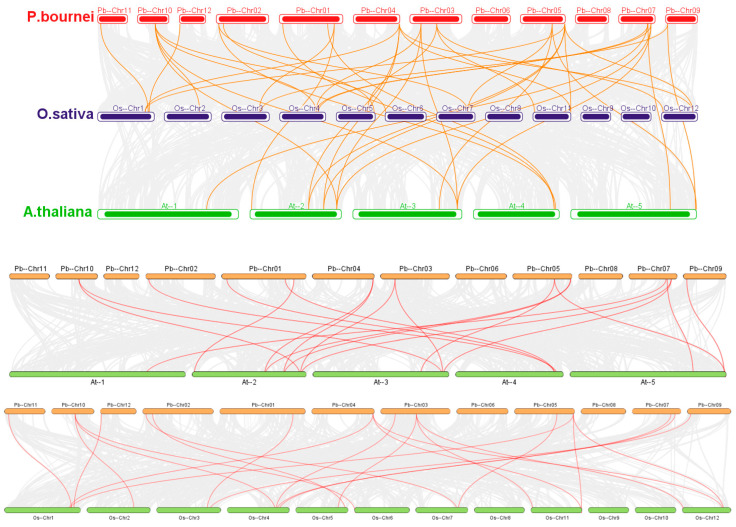
Analysis of homology between the *P. bournei* genome and plant genomes of *A. thaliana* and *O. sativa*. Gray lines symbolize pairs of genomes between homologous blocks, and red lines represent adjacent *PbGRAS* gene pairs. The orange lines emphasize the synthesized GRAS gene pairs in the three species.

**Figure 4 plants-12-02048-f004:**
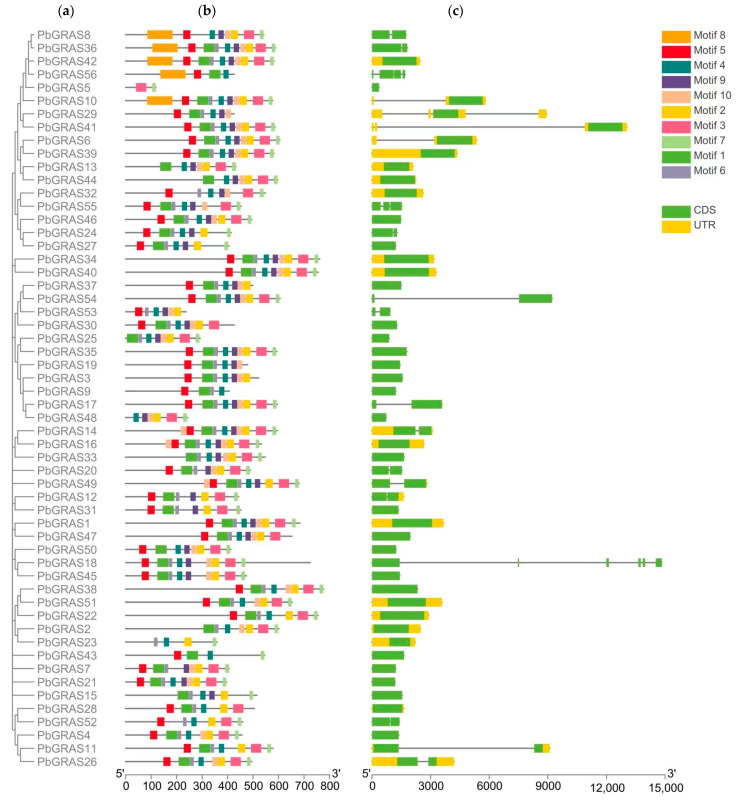
Phylogenetic relationship, motif pattern, and gene structure of *PbGRAS* genes. (**a**) Phylogenetic tree of *PbGRAS*. (**b**) The motifs of PbGRAS. Motifs 1–10 are displayed in different colored boxes. The protein length can be estimated using the scale at the bottom. (**c**) Gene structure of *PbGRAS* genes. Green boxes indicate exons (CDS), black lines indicate introns, and yellow boxes indicate 5′ and 3′ untranslated regions.

**Figure 5 plants-12-02048-f005:**
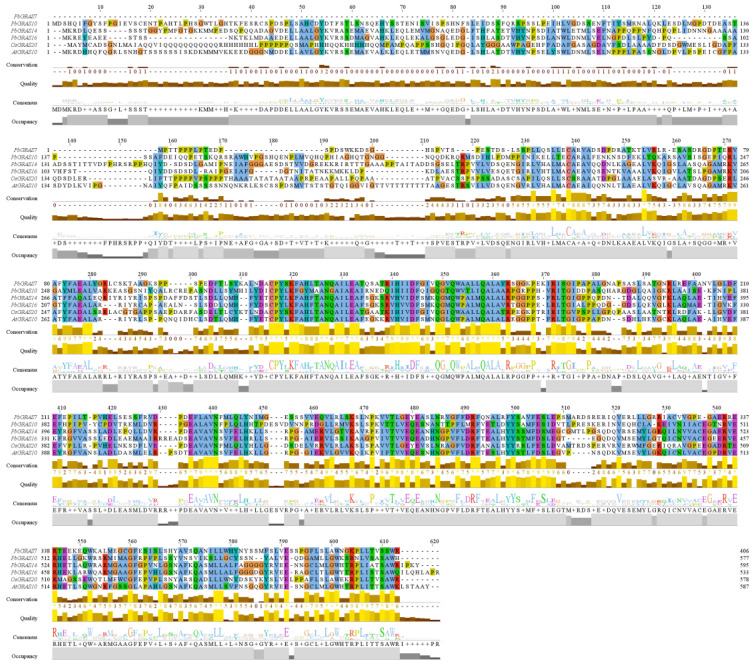
The multiple sequence alignment in *PbGRAS7*, *PbGRAS10*, *PbGRAS14*, *PbGRAS16*, and *PbGRAS40* using Jalview software. The “*” under the conservation in the first line of the annotation indicates the highest similarity, and the number is the output score of the degree of conservatism. In sequence alignment, amino acid alignment of A, I, L, M, F, W, V, C is shown in blue, R, K in red, N, Q, S, T in green, E and D in magenta, G in orange, H, Y in cyan, and P in yellow.

**Figure 6 plants-12-02048-f006:**
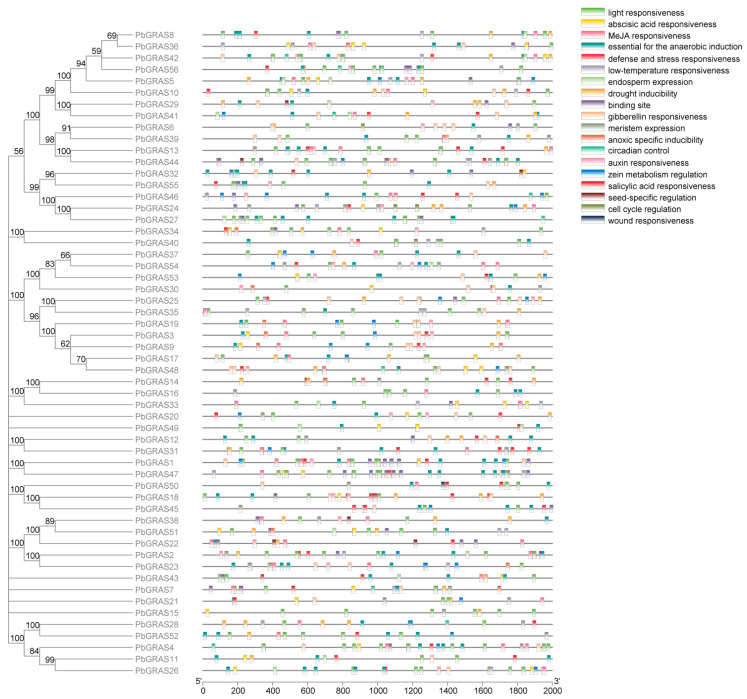
Predicted cis-acting elements in the promoter regions of *PbGRAS* genes. On the left is the ML phylogenetic tree (bootstrap replications: 1000) with branches labeled with bootstrap values. The one on the right is the promoter position (−2000 bp). The cis-acting regulatory elements in the promoter were categorized into 19 types with different colors. The lower axis denotes the quantity of each cis-acting element.

**Figure 7 plants-12-02048-f007:**
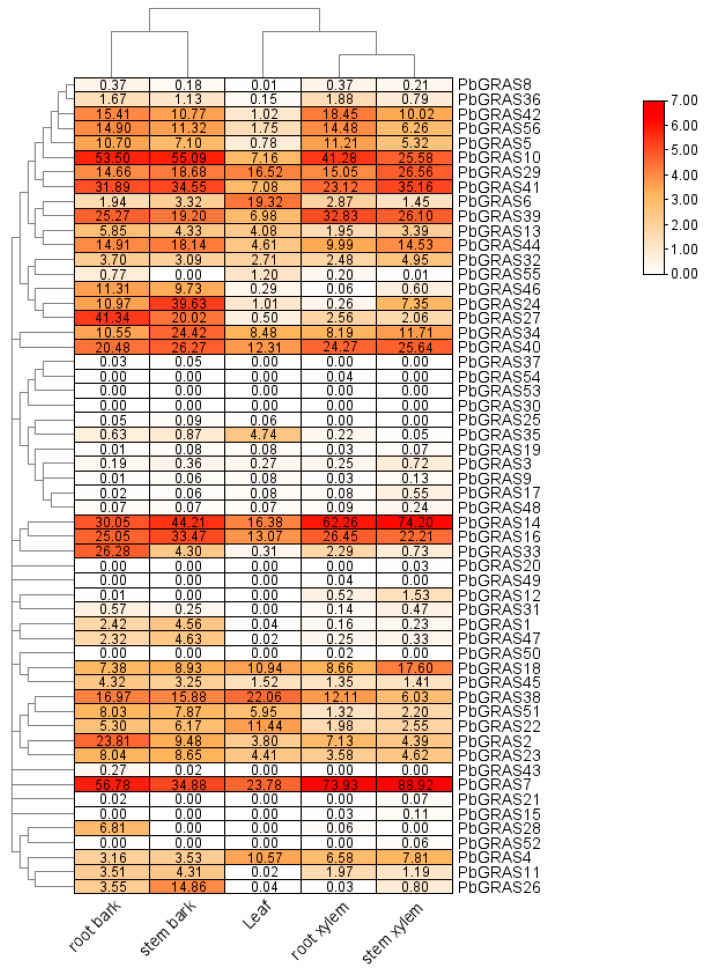
Expression profiling of *PbGRAS*. The expression of *PbGRAS* in five different tissues. The redder the color block, the higher the expression level, and the whiter the color, the lower the expression.

**Figure 8 plants-12-02048-f008:**
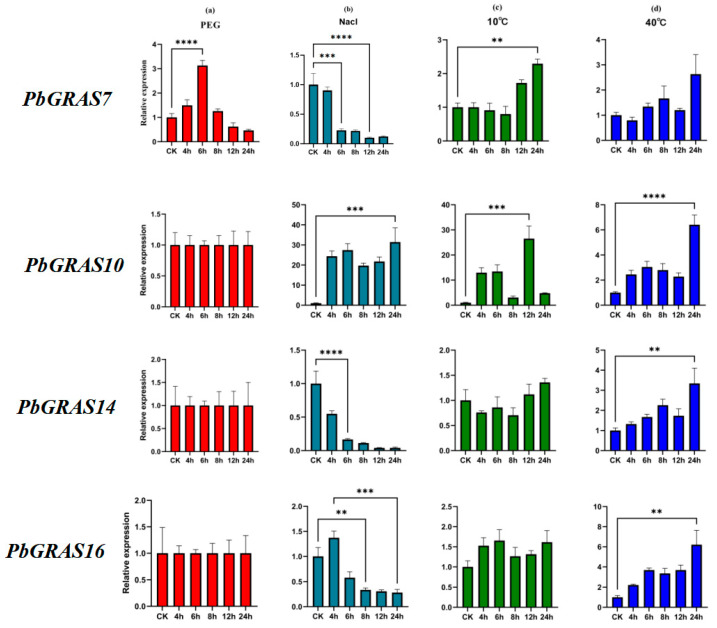
Expression of *PbGRAS* genes in *P. bournei* in response to drought, salt and temperature stresses was tested by means of qRT-PCR. (**a**) Relative gene expression levels at the same time points (4, 6, 8, 12, and 24 h) under treatments with a nutrient solution of 10% PEG simulating a drought environment. The control group was processed in distilled water. (**b**) Relative gene expression levels under the treatment with 10% NaCl nutrient solution immersion. The control group was processed in distilled water. (**c**) Relative gene expression levels at low temperature (10 °C) and for the control (25 °C). (**d**) Relative gene expression levels at high temperature (40 °C) and for the control (25 °C). (** *p* < 0.01, *** *p* < 0.0005, **** *p* < 0.0001).

**Table 1 plants-12-02048-t001:** Characteristics of the GRAS proteins from *P. bournei*.

Member	Number of Amino Acid	Molecular Weight	Theoretical pI	Instability Index	Grand Average of Hydropathicity	Domain (aa)	Intron
*PbGRAS1*	684	76,352.99	5.91	60.92	−0.415	366	0
*PbGRAS2*	601	66,265.46	6.35	61.97	−0.149	364	0
*PbGRAS3*	522	59,304.67	5.94	48.53	−0.263	297	0
*PbGRAS4*	456	49,271.71	5.5	51.05	−0.132	356	0
*PbGRAS5*	119	13,682.77	8.46	39.37	−0.206	116	0
*PbGRAS6*	605	67,162.22	5.54	40.94	−0.315	369	1
*PbGRAS7*	406	44,804.64	5.47	58.55	−0.245	360	0
*PbGRAS8*	542	60,896.42	5.26	55.46	−0.52	329	1
*PbGRAS9*	407	45,766.31	7.25	46.25	−0.329	200	0
*PbGRAS10*	577	64,480.59	5.72	56.43	−0.418	369	1
*PbGRAS11*	578	64,135.67	5.51	51.57	−0.206	351	1
*PbGRAS12*	443	50,201.97	6.11	43.72	−0.368	366	0
*PbGRAS13*	433	48,160.99	9.95	49.35	−0.377	298	0
*PbGRAS14*	595	64,855.31	5.24	45.09	−0.236	366	1
*PbGRAS15*	515	56,248.95	5.28	44.28	−0.229	385	0
*PbGRAS16*	533	58,213.08	5.17	47.33	−0.178	358	0
*PbGRAS17*	595	66,650.92	7.87	47.27	−0.097	374	1
*PbGRAS18*	723	80,619.74	5.5	51.06	−0.284	420	5
*PbGRAS19*	479	54,200.05	6.29	47.41	−0.242	263	0
*PbGRAS20*	489	54,185.93	6.3	53.76	−0.139	345	1
*PbGRAS21*	396	44,773.87	6.19	52.67	−0.24	365	0
*PbGRAS22*	755	82,162.03	5.91	59.13	−0.218	351	0
*PbGRAS23*	359	40,017.22	9.52	60.39	−0.023	341	0
*PbGRAS24*	414	46,394.48	5.14	45.58	−0.259	356	1
*PbGRAS25*	292	33,025.63	7.1	49.73	0.218	278	0
*PbGRAS26*	495	55,471.75	5.63	54.13	−0.385	351	1
*PbGRAS27*	406	45,917.81	5.28	54.2	−0.367	371	0
*PbGRAS28*	505	56,314.09	5.88	42.94	−0.129	354	0
*PbGRAS29*	426	46,516.11	5.27	53.99	−0.293	246	3
*PbGRAS30*	426	48,266.75	5.72	40.08	−0.014	346	0
*PbGRAS31*	452	51,198.4	6.23	47.47	−0.292	369	0
*PbGRAS32*	548	60,019.12	5.69	57.13	−0.52	400	0
*PbGRAS33*	547	59,484.68	5.62	46.5	−0.071	369	0
*PbGRAS34*	762	85,476.64	5.08	50.09	−0.496	372	0
*PbGRAS35*	593	67,285.73	5.14	39.67	−0.224	367	0
*PbGRAS36*	588	65,880.12	5.49	51.14	−0.443	355	0
*PbGRAS37*	499	56,222.09	4.86	35.3	−0.132	276	0
*PbGRAS38*	777	83,607.35	5.81	52.8	−0.168	358	0
*PbGRAS39*	582	64,216.62	5.01	48.6	−0.25	368	0
*PbGRAS40*	756	84,836.7	5.1	47.53	−0.555	372	0
*PbGRAS41*	586	65,363.6	5.7	48.7	−0.304	370	1
*PbGRAS42*	582	65,227.44	5.31	53.36	−0.448	369	0
*PbGRAS43*	545	61,817.03	4.91	46.94	−0.354	369	0
*PbGRAS44*	597	65,108.66	8.49	56.88	−0.344	367	0
*PbGRAS45*	474	53,276.12	5.76	52.93	−0.189	420	0
*PbGRAS46*	495	55,057.59	5.38	53.23	−0.343	381	0
*PbGRAS47*	652	73,122.66	6.4	60.59	−0.443	370	0
*PbGRAS48*	243	27,935.42	6.9	40.59	0.029	240	0
*PbGRAS49*	681	75,504.3	5.71	55.36	−0.333	364	1
*PbGRAS50*	412	46,004.99	7.22	50.54	−0.125	371	0
*PbGRAS51*	653	71,489.74	6.02	54.1	−0.236	363	0
*PbGRAS52*	460	50,659.31	5.91	46.48	−0.223	337	1
*PbGRAS53*	237	26,573	5.9	40.48	0.195	155	1
*PbGRAS54*	606	68,104.02	5.54	32.34	−0.098	372	1
*PbGRAS55*	453	50,294.48	5.66	47.16	−0.077	394	2
*PbGRAS56*	426	47,508.06	4.99	57.1	−0.507	142	2

## Data Availability

Publicly available datasets were analyzed in this study. This data can be found here: The *P. bournei* genome data presented in this study are openly available in CNSA at https://db.cngb.org/search/project/CNP0002030/, accessed on 14 April 2023, CNP0002030. The *A. thaliana* genome data presented in this study are openly available in EnsemblPlents at https://ftp.ensemblgenomes.ebi.ac.uk/pub/plants/release-56/fasta/arabidopsis_thaliana/dna/, accessed on 14 April 2023, Arabidopsis_thaliana.TAIR10.dna.toplevel.fa.gz, 17 February 2023 06:57,35M. The *A. thaliana* genome data presented in this study are openly available in Phytozome v13 at https://phytozome-next.jgi.doe.gov, accessed on 14 April 2023, Osativa_323_v7.0.fa.gz. The *P. bournei* RNA-seq data presented in this study are openly available in https://www.ncbi.nlm.nih.gov/bioproject/, accessed on 14 April 2023, PRJNA628065.

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
