# Peer review of "Genome-Wide Identification, Expression and Stress Analysis of the GRAS Gene Family in *Phoebe bournei"

_plants, 2023, doi:10.3390/plants12102048_

Round 1

Reviewer 1 Report

The work is technically sound piece of research, and within the scope of Plants, it may need a minor revision before its acceptance for publication. 

1. there are some typo/ grammar issues, for example, '10μL cDNA template SYBR' in lane 475. 

2. Provide more references of the year 2023 in the Introduction

May simplify the text via ChatGPT

Reviewer 2 Report

The manuscript submitted by Changet al. describes a genome-wide functional analysis of GRAS transcriptional regulators in Phoebe bournei. As a result, 56 PbGRAS genes were identified, some of which were found to be involved in abiotic stress responses. The topics described are interesting. However, I have found some issues as follows that need to be addressed and corrected.

Figure 5, bottom horizontal line. In my recognition, the promoter sequence should be represented as -2000 to -1.

Line 271-273. The authors should describe why these four genes were selected for further analysis.

Line 278. PEG treatment should be osmotic stress instead of drought stress. And it seems difficult to conclude that PbGRAS7 plays an important role in stress tolerance. The results indicate that PbGRAS7 is involved in the osmotic stress response.

A low temperature-responsive cis-element was predicted at the RbGRAS14 promoter, but its expression level was comparable to control conditions (Figures 5, 7). The authors need to explain a possible hypothesis.

Subfamilies in other species are described in lines 380-391; subfamilies and functions of RbGRAS genes should also be discussed.

The results described in this manuscript provide no direct evidence to support that RbGRAS genes function in resistance to abiotic stresses (line 398-400). These descriptions should be modified.

Minor comments

Line 101. Simply, quantitative RT-PCR (qRT-PCR) should be fine.

Line 118. Figure 3 should be Figure 1.

Line 405. Date should be Data.

The writing style and English of the manuscript has to be checked by a professional editor.

Reviewer 3 Report

Chang et al study the GRAS family in the P.bournei plant species. Nothing is known about this transcription factor family in this genome

The abstract summarizes well their findings.

Introduction

I think the authors need to edit English because in some areas it is a bit hard to understand what they mean and the structure of sentences does not make a lot of sense. GRAS transcription factors are a very versatile family, and a lot has been done so far in many species. Authors need to cadence their paragraphs and include more references about different aspects of the research and from different many more different plants. In addition, since it is such a well-studied as well as important family, authors should include an evolutionary aspect from the perspective of how many genes are in different species.

Results

Figure 1 has no legends for species or groups of GRAS proteins, for the MEGA11 software what other parameters did they use if any?

In their 2.1 section where do they see these results, because the phylogenetic tree is clearly not visible at all!

In section 2.2 authors report that many genes have no introns so I guess the genes are just an exon. It will make sense to check whether those genes might have any transposon elements

In section 2.3 authors report the conservation they should also report the output scores as well as for section 2.4, authors should report the output scores.

I am not sure what Figure 6 reports since in the text (line 254-255) they talk about gene epistasis and in the legend of Figure 6 they report Gene expression. However I only see a heatmap of I am guessing counts…

Materials and Methods

In the 4.1 section authors do not refer to the genome version they used from the public databases

In 4.2 section authors have used blast to find homologs in other species. Given the phylogenetic distance, there are other methods more sensitive like the HMM models. I would strongly suggest authors adopt that method instead.

Based on the Material and Methods I have difficulties to evaluate the Discussion of the paper

Authors need to find a way to correct english.

Round 2

Reviewer 2 Report

With the exception of Figure 5, most of my concerns were addressed and improved in the revised manuscript.

Bottom horizontal line in Figure 5: The authors may have misunderstood my comment. The promoter sequence should be represented as -2000 to -1 in the 5' to 3' direction, as in Figure 4 of this reference paper (doi: 10.3389/fpls.2017.01186.). Furthermore, the transcription start site must be +1 and one base upstream must be -1, so there can be no indication of 0.

Reviewer 3 Report

No comments to make. I accept it
